# The Serine Carboxypeptidase-Like Gene *SCPL41* Negatively Regulates Membrane Lipid Metabolism in *Arabidopsis thaliana*

**DOI:** 10.3390/plants9060696

**Published:** 2020-05-29

**Authors:** Juan Chen, Wei-qi Li, Yan-xia Jia

**Affiliations:** 1Germplasm Bank of Wild Species, Kunming Institute of Botany, Chinese Academy of Sciences, Kunming 650201, Yunnan, China; 2University of Chinese Academy of Science, Beijing 100039, China; chenjuan1@mail.kib.ac.cn

**Keywords:** serine carboxypeptidase-like protein, membrane lipids, phosphatidic acid (PA), 1-butanol

## Abstract

The *Arabidopsis* has 51 proteins annotated as serine carboxypeptidase-like (SCPL) enzymes. Although biochemical and cellular characterization indicates SCPLs involved in protein turnover or processing, little is known about their roles in plant metabolism. In this study, we identified an *Arabidopsis* mutant, *bis4* (*1-butanol insensitive 4*), that was insensitive to the inhibitory effect of 1-butanol on seed germination. We cloned the gene that was defective in *bis4* and found that it encoded an SCPL41 protein. Transgenic *Arabidopsis* plants constitutively expressing *SCPL41* were generated, oil body staining and lipidomic assays indicated that *SCPL41*-overexpressing plants showed a decrease in membrane lipid content, especially digalactosyl diglyceride (DGDG) and monogalactosyl diglyceride (MGDG) contents, while the loss of *SCPL41* increased the membrane lipid levels compared with those in wild-type plants. These findings suggested that *SCPL41* had acquired novel functions in membrane lipid metabolism.

## 1. Introduction

Serine carboxypeptidases (SCPs) and serine carboxypeptidase-like (SCPL) proteins compose a large enzyme family that functions in the hydrolysis of the C-terminal peptide bond in proteins or peptides [1,2,3]. SCPLs share high levels of similarity with SCPs. *Arabidopsis* has at least 51 members [4], while 71 SCPLs exist in the rice genome [1], indicating functional redundancy among members of the SCPL family. Several SCP/SCPL genes have been cloned and studied at the biochemical and/or molecular level from barley [5], wheat [3,6], rice [7,8,9,10,11], tomato [12,13,14], tobacco [15], *Pisum sativum* [16], oats [17], and *Arabidopsis* [18,19,20,21]. Phylogenetic analysis based on protein sequences shows that the acyltransferase of SCPLs has a single origin within the SCPL family [4,17,22]. Their acyltransferase function may have evolved since the divergence of higher plants from mosses, and their genes have undergone independent diversification in monocot and dicot lineages [22]. SCPL proteins in *Arabidopsis* are divided into two major clades. One clade consists of 21 members but only two of them are experimentally characterized as acyltransferases in plants’ secondary metabolism: sinapoylglucose:malate sinapoyltransferase and sinapoylglucose:choline sinapoyltransferase [20]. Another clade comprises 25 members but their biochemical functions are not clearly defined.

Although the functions of most SCPs and SCPLs remain unknown and their known functions are still limited at the biochemical and cellular level, available evidence in plants suggests that they have various functions. For example, *BRS1* (BRI1 Suppressor 1), a member of the SCP family, is a carboxypeptidase [23] working in the brassinosteroid (BR) signaling pathway [18]; SNG1 as acyltransferases involved in the synthesis of sinapoylmalate; glucose acyltransferase which encodes a serine carboxypeptidase-like protein functioned in glucose polyester biosynthesis, IAInos synthase as acyltransferases involved in the synthesis of indole-3-acetic (IAA) ester conjugate [12,22,24,25]; and SNG2 is involved in the biosynthesis of sinapate esters as a sinapoyltransferases [19,26,27]. Other functions include protein turnover and mobilization of storage proteins during seed germination and organ senescence [22,28]. They also have roles in signal transduction [11,18], programmed cell death [29], seed development [3,16], and secondary metabolism for herbivory defense, UV protection, and disease resistance [19,24,27,29].

Major classes of cell membrane lipids include phospholipids and glycolipids. The composition of their molecules greatly affects membrane properties. They also take part in multiple cellular processes, such as signal transduction, cytoskeleton dynamics, vesicle trafficking, and secretion. One important regulatory process is that common membrane phospholipids, such as phosphatidylcholine (PC) and phosphatidylethanolamine (PE), are hydrolyzed by phospholipase D (PLD) to generate a free head-group and phosphatidic acid (PA). PA can serve as a second message to transduce environmental signaling [30,31,32,33,34]. PLD can mediate a specific reaction by catalyzing the transesterification of glycerophospholipids with appropriate small alcohols. For example, PLD catalyzes transphosphatidylation with 1-butanol to produce phosphatidylbutanol [35]. This reaction can block the production of PA and lead to severe physiological consequences. Treatments of seeds with 1-butanol decrease germination and result in growing defective seedlings in *Arabidopsis*. It is, therefore, speculated that PLD-mediated PA is crucial for germination and seedling morphogenesis by maintaining the transverse alignment of microtubules in elongating root cells [35]. However, the mechanism is not fully understood yet.

We identified an *Arabidopsis* mutant, *bis4* (*1-butanol insensitive 4*), that is insensitive to the inhibitory effect of 1-butanol on seed germination. We cloned the gene that is defective in the *bis4* mutant and found that it encodes the SCPL41 protein. SCPL41 protein is a member of clade II of the SCPL family, and its biochemical functions and biological functions have not been clearly defined. We showed that *SCPL41* loss-of-function mutants display PA-insensitive phenotypes in terms of their primary root growth. Transgenic *Arabidopsis* plants constitutively expressing *SCPL41* were generated and found to have lower oil body accumulation. Lipidomic assays indicated that *SCPL41*-overexpressing plants showed a decrease in membrane lipid content, especially digalactosyl diglyceride (DGDG) and monogalactosyl diglyceride (MGDG) content, while the loss of* SCPL41 *increased the membrane lipid levels compared with those in wild-type plants. These findings suggest that *SCPL41* has novel functions in membrane lipid metabolism.

## 2. Results

### 2.1. Identification of the SCPL41 Protein Involved in 1-Butanol Response

We identified a 1-butanol-insensitive mutant, named *bis4* (1-*butanol insensitive 4*), by screening an ethyl methanesulfonate (EMS)-mutagenized plant pool (Col-0 background) for changes in the percentage of seed germination in the presence of exogenous 1-butanol. The mutation site in the mutant was detected (TAIR-PCR) in the seventh exon of *AT5g42230*, which encodes *SCPL41* (serine carboxypeptidase-like gene 41, Figure 1A). The phenotype of *bis44* (SALK_360_G01), an allelic mutant obtained from the *Arabidopsis* Biological Resource Center (ABRC), was comparable to that of *bis4* plants. The germination rates of wild-type, *bis4* and *bis44* mutants were 17.5%, 74.2%, and 58.0% in presence of 0.08% 1-butanol for 4 d, respectively, which suggested germination of the *bis4* and *bis44* mutants both were insensitive to 1-butanol, compared with these processes in wild-type (Appendix A). Compared with *bis44* plant, *bis4* plant was more insensitive to 1-butanol, which might be due to the higher inhibition of *SCPL41* gene expression in *bis4* than in *bis44* plants. The *bis4* line was chosen for subsequent experiments. We generated *SCPL41*-overexpressing plants containing the 35S promoter fused to the *SCPL41* coding region, and two overexpression lines that we named *SOE-5* (*SCPL41* overexpression 5) and *SOE*-7 were identified. The *SCPL41* expression in line 5 was higher than that in line 7, as assessed by qRT-PCR (Figure 1B). The *SOE-5* line was chosen for subsequent experiments.

The *bis4, bis44, SOE-5*, and *SOE-7* plants were compared with the wild-type for their response to 1-butanol during seed germination. The seeds were sown onto media containing different concentrations of 1-butanol for dose–response assay, and the germination percentage was scored when their radicals obviously emerged at 2 d after the end of stratification. In the presence of 1-butanol, seeds without *SCPL41* showed a higher seed germination percentage than did the wild-type, whereas overexpression of *SCPL41* resulted in a significantly reduced seed germination percentage (Figure 1C,D). In the absence of 1-butanol, the seed germination of the different genotypes were comparable. Moreover, in the presence of *tert*-butanol, rather than 1-butanol, the seed germination was comparable to that of the control treatment, and no difference was detected among *Col*, *bis4*, *bis44*, *SOE-5*, and *SOE-7* plants (Figure 1C,D).

The *Col*, *bis4, bis44, SOE*-5*,* and *SOE*-7 plants were also examined for their response to 1-butanol during the postgerminative growth stage. In the absence of 1-butanol or in the presence of *tert*-butanol, cotyledon greening and photochemical quantum efficiency of the photosystem II (PS II) reaction center (F_v_/F_m_) of the three genotypes were comparable (Figure 1E). In the presence of 0.08% 1-butanol, the cotyledon greening and F_v_/F_m_ of *bis4* and *bis44* seedlings were much higher than those of wild-type. In contrast, the cotyledon greening and F_v_/F_m_ of *SOE*-5 and *SOE*-7 were lower than those of the wild-type (Figure 1E,F). In addition, the tested *SCPL41*-overexpression lines revealed that line 5 was more sensitive to 1-butanol than line 7. Given that *SCPL41* expression levels in line 5 were higher than those in line 7 (Figure 1B), our results suggested that the 1-butanol hypersensitivity in these *SCPL41*-overexpression lines is correlated with their *SCPL41* expression levels. Taken together, these results suggested that *SCPL41* acted as a positive regulator of 1-butanol response during germination and postgerminative growth.

To verify whether the different responses of *SCPL41* expression lines to 1-butanol are related to the altered lipid metabolism in these plants, we first compared the total lipid levels in the developing cotyledonary leaves of *Col*, *bis4*, *SOE-5,* and *SOE-7* plants by Nile red staining [36]. The higher the fluorescence intensity, the higher the total lipid content. Significantly lower lipid accumulation was observed in the cotyledons of *SOE-5* and *SOE-7* plants than in those of wild-type plants; however, the total lipid accumulation in the *bis4* mutant was comparable to that in the wild-type plants (Figure 2A,B). These results suggested that *SCPL41* might be involved in lipid metabolism.

### 2.2. SCPL41 Had Different Effects on The Content and Composition of Lipids

Since the above results suggested that *SCPL41* was involved in lipid metabolism, we then comparably profiled the membrane lipids in *Col*, *bis4*, and *SOE-5* plants, and identified >110 molecular species of polar glycerolipids, including six head-group classes of phospholipids, PC, PE, PA, phosphatidylinositol (PI), phosphatidylserine (PS), and phosphatidylglycerol (PG), and two head-group classes of galactolipids, MGDG and DGDG (Table 1). Each molecular species was identified in terms of the total numbers of acyl carbon atoms and double bonds [37].

As an overview, the profile of most lipid species in terms of their absolute level (nmol/mg dry weight, Figure 3, left panel) and their relative content (mol % of total lipids, Figure 3, right panel) was different among wild-type, *bis4,* and *SOE-5 Arabidopsis.* Compared to wild-type, the absolute levels of most lipids were lower in the *SOE-5* plants than in the wild-type *Arabidopsis* but higher in the *bis4* plants (Figure 3). Compared to these in wild-type *Arabidopsis*, the composition of most lipid species in the *SOE-5* plants increased, except for MGDG molecules, while a few lipid species changed in *bis4* plants. These results suggested that the change in *SCPL41* expression dramatically affected lipid degradation and lipid composition, albeit in different ways.

### 2.3. Overexpression of SCPL41 Decreased the Membrane Lipid Content, but Loss of SCPL41 Increased the Membrane Lipid Content

Compared to that in the wild-type, the number of total lipids in *SOE-5* plants decreased to 83.02%, while it increased to 111.91% in *bis4* plants (Table 1). The most important changes concerned the two galactolipids. The level of MGDG in the *SOE-5* plants decreased to 76.15%, and the level of DGDG decreased to 80.58%. The level of PG decreased to 77.57%. The levels of PC and PE were also significantly higher in wild-type plants than in *SOE-5* plants, while no differences in the levels of PI, PA, or PS were detected between wild-type and *SOE-5* plants. However, the deletion of *SCPL41* increased the content of total lipids (from 127.26 nmol/mg to 146.89 nmol/mg, Table 1). Compared to those in the wild-type, the levels of PG, PC, PE, and MGDG in the *bis4* plants increased obviously (Table 1). MGDG 34:6 and 36:6 molecular species as well as the DGDG 36:6 in *SOE-5* plants were much lower than those in wild-type plants (Figure 4). The levels of PG 34:2, 34:3, and 34:4; PC 34:5 and 34:6; and PE 34:2, 34:3, 36:4, and 36:5 were also lower in the *SOE-5* plants than in the wild-type plants. The levels of PC 36:4 and 36:5, PE 34:2 and 34:3 and PG 34:4 increased because of the deletion of *SCPL41* in *bis4* plants. The content of other lipid molecular species showed no clear difference compared with those in wild-type plants (Figure 4). These results indicated that *SCPL41* had a negative correlation between *SCPL41* gene expression and membrane lipid content, further confirming that *SCPL41* regulates membrane lipid metabolism.

### 2.4. SCPL41 Had No Effect on the Composition of Lipid Classes

For the analysis of the relative contents of membrane lipids (mol % total lipids), we only found MGDG and PG percentages slightly lower and PI and PA percentage slightly higher in *SOE-5* plants than that in wild-type (Table 2). The compositions of the lipid molecular species were basically the same among *Col, bis4, and SOE-5* plants (Appendix A). The ratio of galactolipids/phospholipids was 0.74 and 0.65 in wild-type and *SOE-5* plants, respectively, which might have resulted from the relatively low content of MGDG in the *SOE-5* plants (Table 2).

Both PC and DGDG have relatively large head-groups, in terms of chemical structure, and tend to form a bilayer lipid phase. In contrast, PE and MGDG have small head-groups involved in the formation of a nonbilayer lipid phase. The molar PC/PE and DGDG/MGDG ratios are positively related to membrane stability [37,38], we compared these ratios in *bis4* and *SOE-5* plants with that in wild-type plants (Table 2) and found no difference. The degree of unsaturation of membrane lipids significantly impacts the fluidity of the membranes. It is indicated by a double bond index (DBI), an average number of double molecular species. The DBI of each lipid remained unchanged under different levels of *SCPL41* expression (Table 3). These results suggested that *SCPL41* might have no effect on the lipid composition, membrane stabilization, or fluidity.

### 2.5. SCPL41 Had No Effect on the PA content, But Might be Involved in PA Signaling

PA is the product of PLD-mediated phospholipid hydrolysis and plays an important role in many physiological processes [39,40,41]. One process at the seedling stage is that exogenous application inhibits primary root length [42]. We examined the absolute level of PA and the PLD substrate, such as PC and PE, in *Col*, *bis4,* and *SOE-5* plants, and no significant differences in PA levels were detected (Table 1). This suggests that *SCPL41* had no effect on the PA content. However, the exogenous application of PA showed that the inhibition of primary root length is suppressed in *bis4* mutants. The *SOE-5* plants had comparable root length to that of wild-type plants. No obvious difference was detected in the root growth of any of the plants when exposed to exogenous PC (Figure 5). These results suggested that *SCPL41* might play a positive role in PA signaling.

## 3. Discussion

SCP and SCPL family members are expressed in all major tissue types [4] and take part in wide biochemical and cellular processes [16,17,19,24,43,44] in plants, including secondary metabolite biosynthesis, herbicide conjugation, and germination-associated protein degradation in seed reserves. In the present study, we investigate the role of *SCPL41* in lipid metabolism and signaling during the germination and postgerminative growth in *Arabidopsis*. Deletion and overexpression of *SCPL41* confer seeds resistant and sensitive to the inhibitory effect of 1-butanol on seed germination, respectively (Figure 1). The known suppression of 1-butanol on PLD-mediated PA formation led us to investigate the role of *SCPL41* in lipid metabolism. We indeed observed lipids accumulated significantly lower in cotyledonary leaves of *SCPL41* overexpression plants (Figure 2). Lipid profiling further confirmed that deletion and overexpression of *SCPL41* significantly increase and decrease the content of membrane glycerolipids, respectively (Figure 3 and Table 1), while basically maintaining their composition, DBI, and PC/PE and DGDG/MGDG ratios. The PA contents are not affected by *SCPL41* but the inhibitory effect on primary root length is attenuated in the absence of *SCPL41* (Figure 5). Our results demonstrate that *SCPL41* positively regulates membrane lipid metabolism and suggests that it has a positive role in PA signaling.

*SCPL41* was expressed ubiquitously at relatively high levels in flowers but at relatively low levels in seedlings [4]. During the identification and growth of all plant genotypes, no morphological or developmental issue changes were observed, suggesting that *SCPL41* does not have effects on plant growth and development under normal conditions. The increased expression of *SCPL41* caused a large decrease in the contents of PG, MGDG, and DGDG. They are plastidic lipids, the most abundant in leaves [45], and have a direct role in photosynthesis [38]. However, overexpression of *SCPL41* did not affect the lipid composition or the stability of the chloroplast membrane, as reflected by the DGDG/MGDG ratio (Table 2), which also contributed to maintaining photosynthetic activity [46]. This is consistent with the normal photosynthesis activity in *SOE-5* plants, as measured by F_v_/F_m_ (Figure 1E).

SCPLs have been identified by various genetic approaches. Forward genetic screens found mutant plants without specific acylated compounds and led to the identification of *SNG1* [24] and *SNG2* [20] in *Arabidopsis*, and *AsSCPL1* in oat [17]. Activity-guided purification and protein sequencing led to the identification of the tomato GAC enzyme [12]. In our case, *SCPL41* in *Arabidopsis* was identified by screening an EMS mutagenized pool for changes in seed germination in the presence of exogenous 1-butanol (Appendix A). 1-Butanol is involved in the transphosphatidylation reaction as a substrate to produce phosphatidyl alcohol or PA [47]. Thus, we proposed that *SCPL41* might be required for phospholipid metabolism or PA-dependent signaling in *Arabidopsis*. Overexpression of *SCPL41* induced a marked decrease in contents of PC, PE, and PG, which are substrates of phospholipid hydrolysis via PLD, and the deletion of *SCPL41* increased the PC, PE, and PG contents to some extent. The PI and PS contents were not affected when the expression of *SCPL41* changed, which might be because the content was too low to reflect the difference. The SCPL enzymes identified so far are involved in the biosynthesis of glucose polyesters [12], sinapate esters [19,48], anthocyanins, avenacins, tannins [17], and indole-3-acetic acid (IAA) ester conjugates [25,49]. To the best of our knowledge, *SCPL41* is the first member of the SCPL family functioning in cell membrane lipid metabolism.

Previous reports have demonstrated that 1-butanol inhibited seed germination and that 1-butanol blocked PLD-mediated PA produce. It was, therefore, assumed that the 1-butanol inhibition of germination resulted from the absence of PLD-dependent PA [50,51,52]. However, our evidence showed that the insensitivity or hypersensitive of germination to 1-butanol was correlated to the expression of *SCPL41* but the expression of *SCPL41* was not correlated to PA content (Figure 1 and Table 1). Thus, the previous assumption may be incorrect. However, more evidence is needed to prove our speculation. Biochemical characterization of the SCPL41 protein and functional studies of the *SCPL41* gene with knockout/knockdown mutants and/or overexpression in transgenic plants will provide new insights into the biological function and mechanism of *SCPL41* in response to 1-butanol in *Arabidopsis*.

## 4. Material and Methods

### 4.1. Plant Materials and Growth Conditions

*Arabidopsis thaliana* ecotype Columbia (Col-0) was used in the present study. Surface sterilization of all genotype seeds as conducted with 75% ethanol for 2 min followed by 5% NaClO for 2 min and then sterile water rinse three times. The sterilized seeds on MS media with 0.1% agarose were pretreated at 4 °C in darkness for 2 d and then germinated in a growth chamber with conditions of 22 °C, 16-h light/8-h dark photoperiod, and 120 μmol m^−2^ s^−1^ light intensity.

### 4.2. Phenotype Analysis

Seeds of different genotypes were collected from the plants grown in the same conditions at the same time. For phenotype examination, seeds and seedlings (postgerminative growth) of each genotype were germinated and grown side by side on the same plate containing MS medium with different concentrations of 1-butanol and *tert*-butanol as indicated. The germination rate was measured at the percentage of seed with radicle emergence at the indicated times. The percentage of cotyledon greening was recorded after 7 d germination when cotyledons fully expand and turn green. The effect of 1-butanol on cotyledon greening was also quantified by chlorophyll fluorescence of the seedlings (imaging chlorophyll fluorometer, MAXI-Imaging Pulse-Amplitude, Walz/Germany) [53,54] as follows: after dark-adapted for 20 minutes, the maximal quantum yield of photosystem II (PS II) photochemistry (F_v_/F_m_) was measured by the initial fluorescence level (F_0_) and the maximal fluorescence level (F_m_) with the equation of F_v_/F_m_ = (F_m_ − F_0_)/F_m_.

### 4.3. Transgenic Plants

The coding region of *SCPL41* was cloned into a pEGAD-GFP binary vector driven by the cauliflower mosaic virus 35S promoter. *SCPL41*-overexpressing transgenic plants (in the Col-0 background) were generated using Agrobacterium-mediated floral transformation. RT-PCR and resistance screening with glufosinate–ammonium were used to detect the abundance of the transgene transcripts. The homozygous T3 seeds of the transgenic plants were used for analysis.

### 4.4. Nile Red Staining

To visually detect neutral lipids, two-day-old seedlings of wild-type, *bis4*, *SOE-5*, and *SOE-7* were infiltrated with an aqueous solution of Nile red (Sigma) [36,55]. The seedlings were placed in sterile water, cotyledonary leaves were imaged at 10 × and 60 × objective with an Olympus FV 1000 system equipped with argon as an excitation source. Fluorescence was detected at 515 nm and collected with a 539–653 nm filter (*n* = 12 per replicate).

### 4.5. Lipid Analysis and Data Processing

The 7-day-old seedlings were analyzed by electrospray ionization tandem mass spectrometry (ESI-MS/MS) [37] to detect the lipids and the data was processed by the methods previously reported [54]. Lipids in each class were quantified by comparing them with two internal standards and were recorded as nanomoles per milligram of plant dry weight. We analyzed five replicates of each plant species at each sampling time. Paired values were subjected to a *t-*test to determine statistical significance.

The double bond index (DBI) of acyl chains in a given lipid class was calculated using the equation: Double Bond Index (DBI) = [∑(N × mol % fatty acid])/100, where N is the total number of double bonds in each lipid molecule [56].

### 4.6. RNA Isolation and Quantitative PCR

Total RNA was extracted using the RNeasy Plant Mini Kit (Qiagen, QIAGEN N.V. Corporate, Germany) and reverse-transcribed via the reverse transcriptase (Invitrogen, Invitrogen Corporation, United States) according to the manufacturer’s instructions. A total of 50 ng cDNA was used for quantitative PCR in each reaction with SYBR Green PCR Master Mix (Takara, TaKaRa Biotechnology (Dalian) Co.,Ltd., Japan), and then detected by using ABI real-time PCR detection system (Quantstudio 7, QuantStudio 7, Applied Biosystems, United States). Relative amounts of *SCPL41* transcripts were calculated by the comparative cycle threshold method normalized to *ACTIN2* expression from the same sample. Values were determined from three replicates.

## Figures and Tables

**Figure 1 plants-09-00696-f001:**
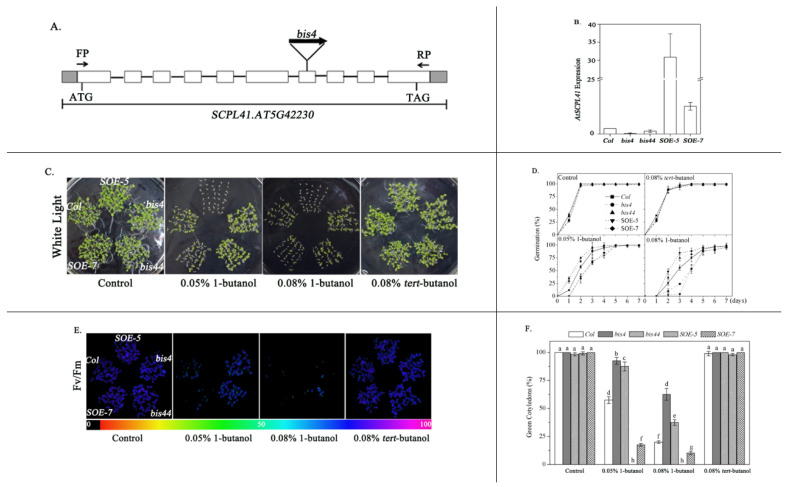
Seed germination and postgerminative growth of *Col, bis4, bis44, SOE-5*, and *SOE-7* plants in response to 1-butanol. (**A**) Gene structure of *AT5g42230*. The filled boxes indicate exons, and the lines indicate introns. The sites of insertion in the *bis4* mutant are marked. (**B**) Quantitative PCR analysis of *SCPL41* expression in 7-d-old *Col*, *bis4*, *bis44*, *SOE-5*, and *SOE-*7 mutant seedlings in the absence of 1-butanol. The values are the means ± SDs (*n* = 3). (**C**) Visual comparison of *Col*, *bis4*, *bis44, SOE-5,* and *SOE-7* seed germination and postgerminative growth after 7 d in the absence or presence of 0.05% 1-butanol, 0.08% 1-butanol, or 0.08% *tert*-butanol. (**D**) Germination rate of *Col* wild-type as well as *bis4*, *bis44*, *SOE-5*, and *SOE-7* plants in the absence or presence of 0.08% 1-butanol or 0.08% *tert*-butanol at the indicated times. (**E**) F_v_/F_m_ values. The color bar on the bottom indicates the F_v_/F_m_ values. (**F**) Green cotyledons were scored after 7 d of growth and represent an average of 120 seeds of at least three independent experiments ± SDs.

**Figure 2 plants-09-00696-f002:**
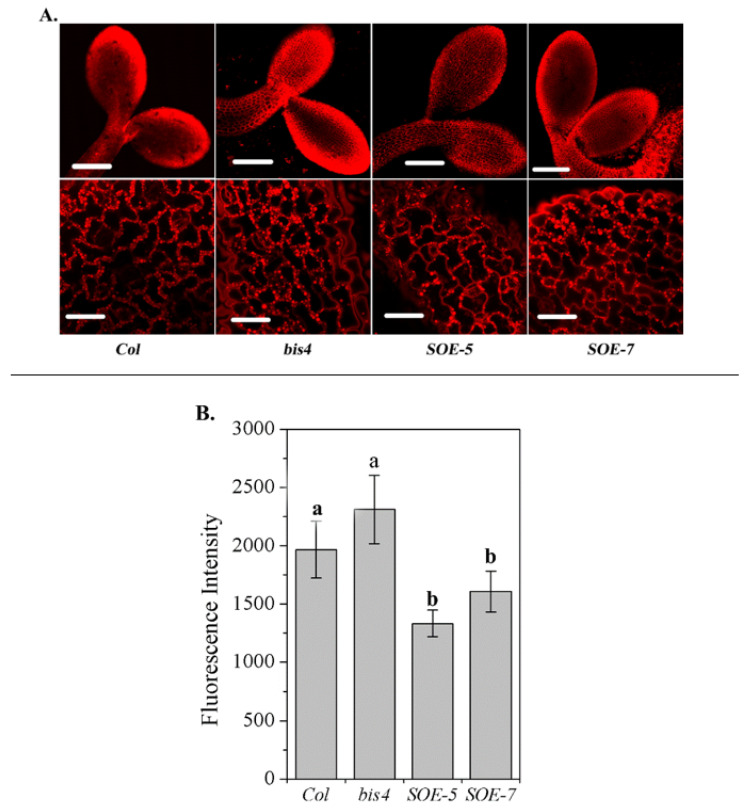
Level of total lipid accumulation in the cotyledonary leaves of *Col, bis4*, *SOE-5, and SOE-7* plants. (**A**) Representative images of lipid accumulation in 2-d-old cotyledonary leaves of *Col*, *bis4* mutant, *SOE-*5, and *SOE-7* plants after staining the seedlings with Nile red. The top panel images were captured at 10 × magnification with an Olympus FV 1000 system. Bars = 200 µm; the bottom panel images were captured at 60 × magnification with an Olympus FV 1000 system. Bars = 30 µm. (**B**) Fluorescence intensity. The average fluorescence intensity of the whole cotyledon was calculated. The higher the fluorescence intensity, the higher the total lipid content.

**Figure 3 plants-09-00696-f003:**
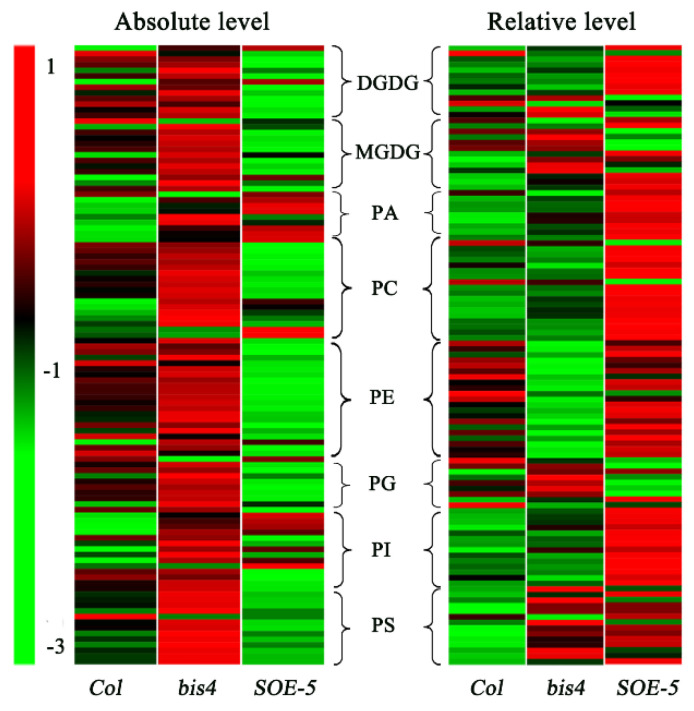
Heat map of lipid molecular species in *Col*, *bis4,* and *SOE-5* plants. Each color bar in the column represents lipid molecular species in the indicated plant. The color of each bar represents the relative change from the median of each lipid species (*n* = 5).

**Figure 4 plants-09-00696-f004:**
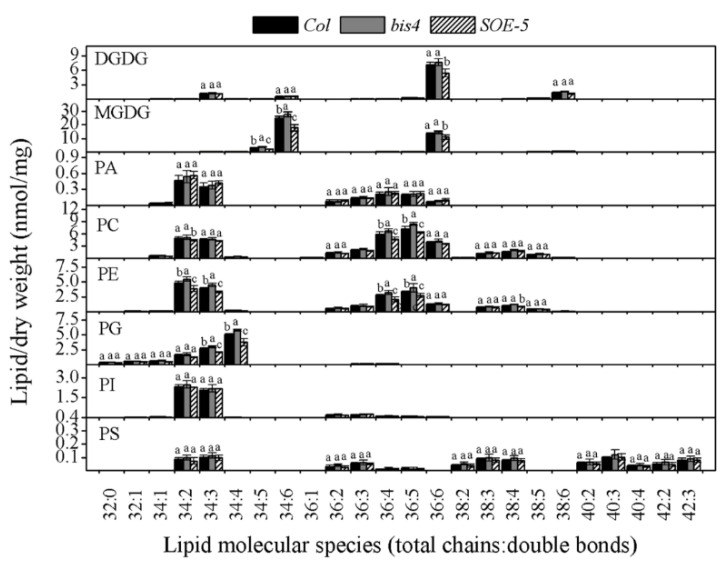
Changes in the molecular species of lipids in *Col, bis4,* and *SOE-5* plants. The values are the means ± SDs (*n* = 4 or 5). Bars for the same lipid class with different letters indicate that the values were significantly different (*P* < 0.05).

**Figure 5 plants-09-00696-f005:**
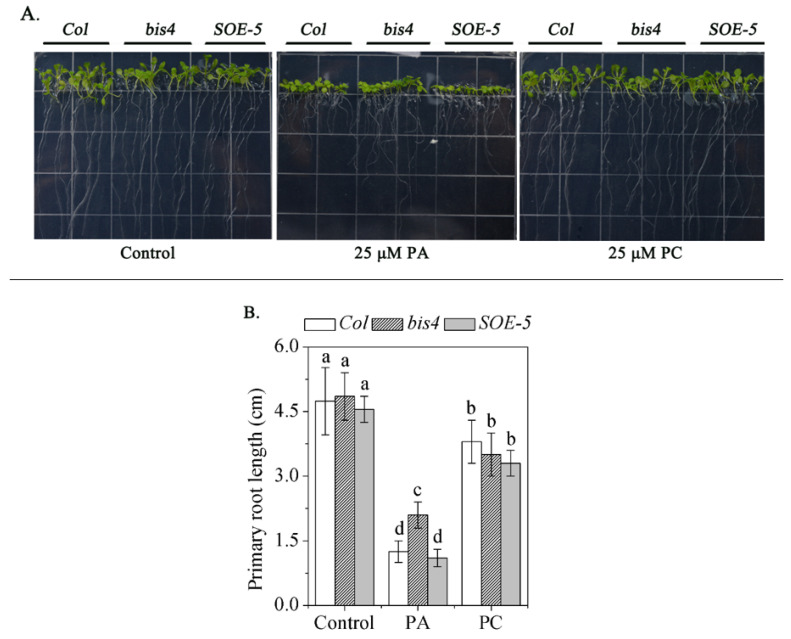
Primary root growth phenotype of *Col*, *bis4,* and *SOE-5* plants. (**A**) Seeds were germinated on MS media for 3 d, after which they were transferred to 25 μM phosphatidylcholine (PC) or phosphatidic acid (PA). (**B**) The length of the primary roots of each genotype plant was averaged after 6 d of vertical growth. Bars with different letters indicate that the value was significantly different (*P* < 0.05).

**Table 1 plants-09-00696-t001:** Total lipids in *Col*, *bis4,* and *SOE-5* plants.

Lipid Classes	Lipids/Dry Weight (nmol/mg)			Ratio	
	*Col*	*bis4*	*SOE-5*	*bis4/Col*	*SOE-5/Col*
PG	10.88 ± 0.45 ^b^	12.28 ± 0.39 ^a^	8.44 ± 0.79 ^c^	112.84	77.57
PC	35.14 ± 2.21 ^b^	38.70 ± 2.44 ^a^	31.43 ± 0.45 ^c^	110.13	89.43
PE	21.02 ± 0.95 ^b^	23.85 ± 1.33 ^a^	17.41 ± 1.72 ^c^	113.46	82.87
PI	5.44 ± 0.32 ^a^	5.79 ± 0.69 ^a^	5.53 ± 0.17 ^a^	-	-
PA	1.55 ± 0.25 ^a^	1.74 ± 0.34 ^a^	1.81 ± 0.15 ^a^	-	-
PS	0.88 ± 0.06 ^a^	1.02 ± 0.25 ^a^	0.80 ± 0.17 ^a^	-	-
MGDG	44.33 ± 2.11 ^b^	49.43 ± 2.76 ^a^	33.75 ± 4.54 ^c^	111.50	76.15
DGDG	11.42 ± 0.80 ^a^	12.09 ± 1.08 ^a^	9.20 ± 1.09 ^b^	-	80.58
	**Total lipids/dry weight (nmol/mg)**				
Total lipids	127.26 ± 2.37 ^b^	146.89 ± 8.11 ^a^	108.97 ± 7.65 ^c^	115.43	83.02

*Col, bis4 and SOE-5* plants with different letters a, b and c in the same row indicate significantly differences. (*P* < 0.05). The values are the means ± SDs (*n* = 5).

**Table 2 plants-09-00696-t002:** Membrane lipid composition in each head-group class and lipid ratio in *Col*, *bis4,* and *SOE-5* plants.

Lipid Classes	Lipid (mol % of total lipids)		
	*Col*	*bis4*	*SOE-5*
PG	8.41 ± 0.31 ^a^	8.61 ± 0.38 ^a^	7.85 ± 0.26 ^b^
PC	26.72 ± 0.74 ^a^	26.79 ± 0.99 ^a^	28.91 ± 2.12 ^a^
PE	16.01 ± 0.42 ^a^	14.47 ± 2.72 ^a^	15.98 ± 0.99 ^a^
PI	4.14 ± 0.13 ^b^	4.24 ± 0.34 ^ab^	5.01 ± 0.41 ^a^
PA	1.18 ± 0.19 ^b^	1.28 ± 0.23 ^b^	1.66 ± 0.11 ^a^
PS	0.67 ± 0.04 ^a^	0.74 ± 0.11 ^a^	0.73 ± 0.14 ^a^
MGDG	33.72 ± 0.36 ^a^	34.55 ± 1.25 ^a^	30.81 ± 2.14 ^b^
DGDG	8.70 ± 0.31 ^a^	8.88 ± 0.31 ^a^	8.42 ± 0.43 ^a^
	**Lipid ratio**		
PC/PE	1.67 ± 0.06 ^a^	1.93 ± 0.53 ^a^	1.82 ± 0.18 ^a^
DGDG/MGDG	0.26 ± 0.01 ^a^	0.26 ± 0.01 ^a^	0.27 ± 0.01 ^a^
Galactolipids/Phospholipids	0.74 ± 0.02 ^a^	0.77 ± 0.04 ^a^	0.65 ± 0.07 ^b^

*Col, bis4 and SOE-5* plants with different letters a, b and c in the same row indicate significantly differences. (*P* < 0.05). The values are the means ± SDs (*n* = 4 or 5).

**Table 3 plants-09-00696-t003:** Double bond index (DBI) of each lipid class.

Lipid Class	DBI (double bond index)		
	*Col*	*bis4*	*SOE-5*
DGDG	6.48 ± 0.02 ^b^	6.51 ± 0.01 ^a^	6.39 ± 0.04 ^c^
MGDG	6.84 ± 0.01 ^a^	6.84 ± 0.01 ^a^	6.83 ± 0.01 ^a^
PG	2.98 ± 0.04 ^a^	2.97 ± 0.04 ^a^	2.94 ± 0.07 ^a^
PC	3.81 ± 0.02 ^a^	3.82 ± 0.02 ^a^	3.80 ± 0.01 ^a^
PE	3.43 ± 0.02 ^a^	3.44 ± 0.02 ^a^	3.43 ± 0.02 ^a^
PI	2.59 ± 0.01 ^a^	2.58 ± 0.01 ^a^	2.58 ± 0.01 ^a^
PS	2.70 ± 0.02 ^a^	2.69 ± 0.05 ^a^	2.75 ± 0.04 ^a^
PA	3.06 ± 0.02 ^a^	3.11 ± 0.06 ^a^	3.10 ± 0.09 ^a^

*Col, bis4 and SOE-5* plants with different letters a, b and c in the same row indicate significantly differences. (*P* < 0.05). The values are the means ± SDs (*n* =4 or 5).

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
