# Peer review of "The Serine Carboxypeptidase-Like Gene SCPL41 Negatively Regulates Membrane Lipid Metabolism in Arabidopsis thaliana"

_plants, 2020, doi:10.3390/plants9060696_

Round 1

Reviewer 1 Report

This paper by Chen et al. demonstrates function of Serine Carboxypeptidase-like Gene SCPL41 in membrane lipid metabolism. In general, the experiments are performed carefully, with the appropriate controls, and the data support the conclusions drawn. The manuscript is well-written and organized and I would recommend publication if authors address the points described below.

1)  It is difficult to understand whether bis4 was identified in this study or in the previous study. In the abstract it is mentioned that bis4 was identified previously and then in the introduction (L71) it is stated, “We identified an Arabidopsis mutant, bis4…”, sounds like it is identified in this study. If previously identified, please provide the reference.

2) Figure 1: Subfigure labels are missing

3) Figure 1A: SCPL41. AT5G10290 - Wrong accession is used (leucine-rich repeat transmembrane protein kinase family protein). Please correct.

4) Figure 1B: Could you please clarify that a partial protein sequence of SCPL41 in the bis4 mutant (from start codon till the site of insertion) synthesized doesn’t cause a dominant negative effect in the plant?

5) Figure 2. It’s difficult to judge the level of total lipid accumulation in the cotyledonary leaves in the microscopic image of Nile Red Staining. I think an additional comparative bar graph would be useful.

6) The bis4 mutant is insensitive to the inhibitory effect of 1-butanol on seed germination as well as post germinative growth stage because of the role of SCPL41 in membrane lipid metabolism. It is therefore critical to study how, and to what extent 1-butanol affects lipid metabolism in SCPL41 mutants and overexpessor lines. However, except for the initial growth experiments, 1-butanol was not used in other experiments. Could you please clarify this?

Author Response

This paper by Chen et al. demonstrates function of Serine Carboxypeptidase-like Gene SCPL41 in membrane lipid metabolism. In general, the experiments are performed carefully, with the appropriate controls, and the data support the conclusions drawn. The manuscript is well-written and organized and I would recommend publication if authors address the points described below.

1) It is difficult to understand whether bis4 was identified in this study or in the previous study. In the abstract it is mentioned that bis4 was identified previously and then in the introduction (L71) it is stated, “We identified an Arabidopsis mutant, bis4…”, sounds like it is identified in this study. If previously identified, please provide the reference.

Response 1: The bis4 mutant was identified in this study. There was a mistake in the sentence of Abstract and the wrong word “previous” is deleted (line 12-13). We are sorry for this confusion.

2) Figure 1: Subfigure labels are missing

Response 2: The subfigure labels have been added in new version of manuscript (lines 94-95, lines 122-127, lines 146-147, lines 236-237).

3) Figure 1A: SCPL41. AT5G10290 - Wrong accession is used (leucine-rich repeat transmembrane protein kinase family protein). Please correct.

Response 3: This is a mistake and has been corrected in the new version of manuscript (line 123).

4) Figure 1B: Could you please clarify that a partial protein sequence of SCPL41 in the bis4mutant (from start codon till the site of insertion) synthesized doesn’t cause a dominant negative effect in the plant?

Response 4: We indeed compared the growth and development between WT and mutants but found no difference under normal growth conditions. It could be due to function redundancy among SCPL family members, or that biochemical effects of SCPL41 is not strong enough to cause morphological and/or developmental malformation under normal conditions, considering that no physiological phenotype associated SCPL41 is reported so far. We add sentences to clarify this issue (lines 260-261).

5) Figure 2. It’s difficult to judge the level of total lipid accumulation in the cotyledonary leaves in the microscopic image of Nile Red Staining. I think an additional comparative bar graph would be useful.

Response 5: This is a very good suggestion. We add additional comparative bar graph as suggested and calculate the fluorescence intensity to characterize the total lipid contents (lines 143-149).

6) The bis4 mutant is insensitive to the inhibitory effect of 1-butanol on seed germination as well as post germinative growth stage because of the role of SCPL41 in membrane lipid metabolism. It is therefore critical to study how, and to what extent 1-butanol affects lipid metabolism in SCPL41 mutants and overexpessor lines. However, except for the initial growth experiments, 1-butanol was not used in other experiments. Could you please clarify this?

Response 6: This is a good point. We add sentences to clarify this question in Discussion section (lines 282-291). Previous reports showed that 1-butanol inhibited seed germination/post germinative growth and that 1-butanol blocked PLD-mediated PA produce (by exogenous application of 1-butanol and in vitro measurements). It was therefore assumed that the 1-butanol inhibition of germination resulted from the absence of PLD-dependent PA. However, our evidence (genetic and in vivo measurements) showed that the insensitivity or hypersensitive of germination to 1-butanol was correlated to the expression of SCPL41 but the expression of SCPL41 was not correlated to PA content (Figure 1 and Table 1). Thus, the previous assumption may be incorrect. We are actually working on the molecular mechanism of this tissue. However, we have to say that this is a complicate and big issue, involved in hormone signaling and protein interaction, and is not finished yet. It is better for us to separate whole set of data into two stories. We feel that the present data is mainly about the role of SCPL41 in lipid metabolism and could form an independent story. The questions you mentioned would be addressed in another story in future.

Reviewer 2 Report

The article entitled “The Serine Carboxypeptidase-like Gene SCPL41 3 Negatively Regulates Membrane Lipid Metabolism 4 in Arabidopsis thaliana” by Chen et al., described the role of SCPL41 in lipid metabolism. The authors used different studies including inhibition of function as well as overexpression of SCPL41 protein to prove their hypothesis. The authors had earlier identified an Arabidopsis mutant, bis4 that was insensitive to the inhibitory effect of 1-buatnol on seed germination. The authors cloned the gene which was supposed to be defective in bis4 and it was reported to encode for SCPL41 protein. The authors generated two transgenic lines that constitutively express SCPL41. The authors showed that these transgenic lines showed decrease in membrane lipid content. The authors further showed that when there is loss of SCPL41 the level of membrane lipid increased when compared with that of in wild type. The authors described that SCPL41 is required for normal seed germination in response to 1-butanol. In Figure S1b legend, the authors mentioned that the germination results are shown in the absence or presence of 0.05%, 0.08% 1-butanol while in graphs they only showed for 0.08% 1-butanol. Similarly, in text the authors mentioned about the germination percent after 4 days (page 2, line 90) while in figure S1 they showed graph of week. The authors should either present germination percent of every day or after day 7. Further, while demonstrating the level of total lipid accumulation in cotyledonary leaves of wild type and mutant plants in Figure 2, the authors used Nile red staining. It appears that there is no difference in expression level. The authors should measure the expression level and provide it graphically. The authors should perform Nile red staining for bis44 mutant plants too. The authors require to present cluster analysis for SOE-7 plants too as it will provide readers an insight of what’s happening between two transgenic lines which is having similar target. The authors should provide valid explanation for the reason of difference in expression of various lipid synthesis genes in SOE-5 and SOE-7 transgenic lines. In last, there are few places where authors require to rewrite the sentences for better clarity such as Page 1, line 37. The authors should go through the manuscript once more and check for any other sentence that might require rewriting for better clarity.

Author Response

The article entitled “The Serine Carboxypeptidase-like Gene SCPL41 Negatively Regulates Membrane Lipid Metabolism in Arabidopsis thaliana” by Chen et al., described the role of SCPL41 in lipid metabolism. The authors used different studies including inhibition of function as well as overexpression of SCPL41 protein to prove their hypothesis. The authors had earlier identified an Arabidopsis mutant, bis4 that was insensitive to the inhibitory effect of 1-buatnol on seed germination. The authors cloned the gene which was supposed to be defective in bis4 and it was reported to encode for SCPL41 protein. The authors generated two transgenic lines that constitutively express SCPL41. The authors showed that these transgenic lines showed decrease in membrane lipid content. The authors further showed that when there is loss of SCPL41 the level of membrane lipid increased when compared with that of in wild type. The authors described that SCPL41 is required for normal seed germination in response to 1-butanol.

In Figure S1b legend, the authors mentioned that the germination results are shown in the absence or presence of 0.05%, 0.08% 1-butanol while in graphs they only showed for 0.08% 1-butanol.

Response 1: The sentences and the figures have been revised (please see lines 94-95, lines 124-128).

Similarly, in text the authors mentioned about the germination percent after 4 days (page 2, line 90) while in figure S1 they showed graph of week. The authors should either present germination percent of every day or after day 7.

Response 2: The most significant difference of germination among genotypes exhibited after 4 days treatment under 0.08% 1-butanol. We therefore describe the germination rate on the fourth day in the text.

Further, while demonstrating the level of total lipid accumulation in cotyledonary leaves of wild type and mutant plants in Figure 2, the authors used Nile red staining. It appears that there is no difference in expression level. The authors should measure the expression level and provide it graphically. The authors should perform Nile red staining for bis44 mutant plants too.

Response 3: We add additional comparative bar graph as Reviewer 1 suggested, and calculate the fluorescence intensity to characterize the total lipids content (lines 143-147, please also see Response 5).

The authors require to present cluster analysis for SOE-7 plants too as it will provide readers an insight of what’s happening between two transgenic lines which is having similar target.

Response 4: After the phenotypes of two function-loss mutants bis4 and bis44 and two overexpression lines of soe-5 and soe-7 were verified, we chose the most representative mutant (bis4) and overexpression line (SOE-5), in terms of both gene expression and physiological behavior, to conduct subsequent lipidomics measurement. The lipid data of SOE-5 line can provide a significant insight because 1) it is consistent with the lipid data by Nile staining, in which both SOE-5 and SOE-7 lines are employed, and 2) it is reasonable in comparison with those of bis4 mutant.

The authors should provide valid explanation for the reason of difference in expression of various lipid synthesis genes in SOE-5 and SOE-7 transgenic lines.

Response 5: The expression data of lipid synthesis genes is removed. The expression of lipid synthesis genes in different genotypes has no significant correlation with SCPL41 expression. It does not contribute to elucidate the role of SCPL41 gene in lipid metabolism. We realize that it is redundant for the bio-context and beyond the focus of this study.

In last, there are few places where authors require to rewrite the sentences for better clarity such as Page 1, line 37. The authors should go through the manuscript once more and check for any other sentence that might require rewriting for better clarity. 

Response 6: The sentences have been checked and revised. We thank Reviewer 3 for his/her careful reading.

Reviewer 3 Report

The work by Juan-Chen et al. investigates the serine carboxypeptidase-like (SCPL) enzymes function in Arabidopsis mutant. To address physiological role one of these proteins, the SCPL41 protein, authors both performed a screen for defective strains and generated transgenic plants constitutively over-expressing SCPL41. Phenotypic characterization coupled with lipidomic approaches allowed the authors to correlate the membrane lipid content with the SCPL41 expression level.

While the purpose to consider the SCPLs biological function is exciting and the work is properly performed, the paper would certainly benefit from a few rearrangements.

  1. I think that the quality of the written text would require some improvements. First, it is not clear where the work goes beyond a potential previous report that already characterized the bis4 mutant. This point (“We previously identified an Arabidopsis mutant, bis4 (1-butanol insensitive 4),” Line 14) should be clarified. I also would encourage authors to respect more finely the arrangement chosen for the different paragraphs. Namely, the results section shows a continuous exchange of ideas... This is not helpful for the reader. I believe that it would be better to comment the results successively and independently, rather than continuously report to all the data…

  1. I would also advice to correct some ambiguities:

2.1. I would maybe change “several” to “three” (Line 37) and “similar” to “comparable” (Line 89) for more accuracy.

2.2. I regret that the distinction between total vs membrane lipids may not always be obvious. They are some confusing allegations. For example, the whole paragraph from line 184 to line 191 correctly refers to total lipids and in agreement to Table 1. However, the conclusion mentions “membrane lipids” (Line 193), which are precisely described below (Lines 218-224). I would recommend careful editing and proofreading of this point.

2.3. “These results indicated that the change in SCPL41 expression affected the content of lipids derived from the eukaryotic pathway and the prokaryotic pathway but may not be related to the synthesis of lipids” (Lines 203-205). No convincing evidence was presented to support this statement. Thus, the authors need to propose supplemental data to support their conclusion or to be more cautious.

2.4. These results indicated that SCPL41 had no effect on the composition of lipids, cell membrane stabilization or fluidity (Lines 236-237). Similarly, without results strictly characterizing the biophysical state of the membrane, I would recommend to be more cautious. I agree with the consideration that “the degree of unsaturation of membrane lipids significantly impacts the fluidity of the membranes” (Lines 231-232). Nevertheless, this parameter is clearly not a direct measurement of membrane stability and/or fluidity. FRAP experiments, electrolyte leakage measurements, … can for example help to validate such strong assessment.

2.5. “SCPL41 was expressed ubiquitously at relatively high levels in flowers but at relatively low levels in seedlings [4]. In our studies, neither overexpression of SCPL41 or deletion of SCPL41 caused any morphological or developmental issue changes, at least in seedlings, suggesting that SCPL41 might have limited function in plant growth and development.” (Lines 280-284). Even then, I have personally greater reservations about this sentence. Since the expression level is low in seedling, I would think that it is not the best organ to evaluate the protein involvement on growth and development. The conclusion could then be excessive.

2.6 I would refer to figure 1B (Line 94) when analysing bis4 and bis44 SCPL41 expression level, and, as above, I would add “The bis4 line was chosen for subsequent experiments“, in this part of the text.

  1. The figures could be easily improved.

3.1. Figure S1 – Different values are reported in legend and figure (respectively, 0.08 and 0.05% of tert-butanol), moreover, 0.05% is indicated in the legend but the corresponding panel missed in the figure. In the legend, “A” and “B” items are mentioned, I would suggest adding them in the figure to help build a greater consistency.

3.2. Figure 1 – Similarly, I would propose to annex “A”, “B”, … items in the figure. Distinct panels could also be reorganized to follow the logic of the draft text and make the readers much less confusing (for example, putting back the photography reporting the Fv/Fm data close to the cotyledons colours measurements).

3.3. Figure 2 – The idea to qualitatively observe lipid accumulation is quite interesting. However, I would recommend to the authors to propose a more defined figure (especially for SOE-7) to be consistent with the sentence “Significantly lower lipid accumulation was observed in the cotyledons of SOE-5 and SOE-7 plants than in those of wild-type plants; …” (Lines 131-133). I unambiguously agree only with the data corresponding to SOE-5.

3.4. Figure 3 – Such data are generally difficult to interpret. I observed similar shades of red (MGDG) or green (PE) between the two panels. I would submit to change colours scale to avoid misinterpretation of this figure and to fully conform to the sentences “The absolute levels of most lipids were lower in the SOE-5 plants than in the wild-type Arabidopsis but higher in the bis4 plants (Figure 3). Compared to these in wild-type Arabidopsis, the composition of most lipid species in the SOE-5 plants increased, except for MGDG molecules, while no obvious difference was detected between the wild-type and bis4 plants.” (Lines 168-172).

  1. Minor typing errors could be corrected. For example, in the abstract “SCPL” abbreviation is done two times; I also assume that some sentences are not properly written (Lines 37-40, I would suggest to remove “that”; Line 218, I wonder if I would not start with “from” rather than with “For”, …).

Author Response

The work by Juan-Chen et al. investigates the serine carboxypeptidase-like (SCPL) enzymes function in Arabidopsis mutant. To address physiological role one of these proteins, the SCPL41 protein, authors both performed a screen for defective strains and generated transgenic plants constitutively over-expressing SCPL41. Phenotypic characterization coupled with lipidomic approaches allowed the authors to correlate the membrane lipid content with the SCPL41expression level.

While the purpose to consider the SCPLs biological function is exciting and the work is properly performed, the paper would certainly benefit from a few rearrangements.

  1. I think that the quality of the written text would require some improvements. First, it is not clear where the work goes beyond a potential previous report that already characterized the bis4 mutant. This point (“We previously identified an Arabidopsis mutant, bis4 (1-butanol insensitive 4),” Line 14) should be clarified. I also would encourage authors to respect more finely the arrangement chosen for the different paragraphs. Namely, the results section shows a continuous exchange of ideas. This is not helpful for the reader. I believe that it would be better to comment the results successively and independently, rather than continuously report to all the data…

Response 1: The bis4 mutant was identified in this study. The wrong word is deleted (lines 12-13, please also see Response 1). The result section has been revised as indicated below. We appreciate Reviewer 2’s suggestions.

2. I would also advice to correct some ambiguities:

2.1. I would maybe change “several” to “three” (Line 37) and “similar” to “comparable” (Line 89) for more accuracy.

Response 2: The sentences have been revised in new version of manuscript as suggested (lines 35, 108-109, 145, and 233).

2.2. I regret that the distinction between total vs membrane lipids may not always be obvious. They are some confusing allegations. For example, the whole paragraph from line 184 to line 191 correctly refers to total lipids and in agreement to Table 1. However, the conclusion mentions “membrane lipids” (Line 193), which are precisely described below (Lines 218-224). I would recommend careful editing and proofreading of this point.

Response 3: This part has been modified (please see lines 178-218). We appreciate Reviewer 2’s suggestions.

2.3. “These results indicated that the change in SCPL41 expression affected the content of lipids derived from the eukaryotic pathway and the prokaryotic pathway but may not be related to the synthesis of lipids” (Lines 203-205). No convincing evidence was presented to support this statement. Thus, the authors need to propose supplemental data to support their conclusion or to be more cautious.

Response 4: The sentences involved in “eukaryotic pathway and the prokaryotic pathway” are removed (please see lines 178-195).

2.4. These results indicated that SCPL41 had no effect on the composition of lipids, cell membrane stabilization or fluidity (Lines 236-237). Similarly, without results strictly characterizing the biophysical state of the membrane, I would recommend to be more cautious. I agree with the consideration that “the degree of unsaturation of membrane lipids significantly impacts the fluidity of the membranes” (Lines 231-232). Nevertheless, this parameter is clearly not a direct measurement of membrane stability and/or fluidity. FRAP experiments, electrolyte leakage measurements, … can for example help to validate such strong assessment.

Response 5: We agree Reviewer 2 with that our evidence about the biophysical properties of membranes is indirect. We modify this paragraph and soften the  description of our conclusion (lines 205-218).

2.5. “SCPL41 was expressed ubiquitously at relatively high levels in flowers but at relatively low levels in seedlings [4]. In our studies, neither overexpression of SCPL41 or deletion of SCPL41 caused any morphological or developmental issue changes, at least in seedlings, suggesting that SCPL41 might have limited function in plant growth and development.” (Lines 280-284). Even then, I have personally greater reservations about this sentence. Since the expression level is low in seedling, I would think that it is not the best organ to evaluate the protein involvement on growth and development. The conclusion could then be excessive.

Response 6: It is true that SCPL41 may affect morphology and development at late growth stage and under stress conditions. We modify these sentences and add words about our observation on plant growth (lines 243-293). During our work in mutant identification and growth of all genotype plants under normal conditions, no morphological or developmental issue was observed. Please also see Response 4.

2.6 I would refer to figure 1B (Line 94) when analysing bis4 and bis44 SCPL41 expression level, and, as above, I would add “The bis4 line was chosen for subsequent experiments“, in this part of the text.

Response 7: We add “The bis4 line was chosen for subsequent experiments” in this part of the text (please see lines 88-89). We thank Reviewer 2 for this good suggestion.

3. The figures could be easily improved.

3.1. Figure S1 – Different values are reported in legend and figure (respectively, 0.08 and 0.05% of tert-butanol), moreover, 0.05% is indicated in the legend but the corresponding panel missed in the figure. In the legend, “A” and “B” items are mentioned, I would suggest adding them in the figure to help build a greater consistency.

Response 8: The sentences and the Figures have been revised and relevant words have been added in new version of manuscript (please see lines 94-95).

3.2. Figure 1 – Similarly, I would propose to annex “A”, “B”, … items in the figure. Distinct panels could also be reorganized to follow the logic of the draft text and make the readers much less confusing (for example, putting back the photography reporting the Fv/Fm data close to the cotyledons colours measurements).

Response 9: Sorry for these omissions. The figures have been revised according to the reviewer’s advice (please see lines 122-125 and 126-127).

3.3. Figure 2 – The idea to qualitatively observe lipid accumulation is quite interesting. However, I would recommend to the authors to propose a more defined figure (especially for SOE-7) to be consistent with the sentence “Significantly lower lipid accumulation was observed in the cotyledons of SOE-5 and SOE-7 plants than in those of wild-type plants; …” (Lines 131-133). I unambiguously agree only with the data corresponding to SOE-5.

Response 10: According to the review’s suggestion, we add additional comparative bar graph and calculate the fluorescence intensity to characterize the total lipids content (lines 144-147 and also see Response 5).

3.4. Figure 3 – Such data are generally difficult to interpret. I observed similar shades of red (MGDG) or green (PE) between the two panels. I would submit to change colours scale to avoid misinterpretation of this figure and to fully conform to the sentences “The absolute levels of most lipids were lower in the SOE-5 plants than in the wild-type Arabidopsis but higher in the bis4 plants (Figure 3). Compared to these in wild-type Arabidopsis, the composition of most lipid species in the SOE-5 plants increased, except for MGDG molecules, while no obvious difference was detected between the wild-type and bis4 plants.” (Lines 168-172).

Response 11: The Figure 3 in the last manuscript showed the original data with five replicates in each genotype. We realize that this form of data does not really reflect the changes of lipid content. We replace it with the form of average value of the five independent replicates. The data is subjected to Q-test before average calculation (Figure 3 of new version) as the previous reported. We believe that the overview in new version of Figure3 can show the difference between genotypes in terms of both absolute level (nmol/mg DW) and relative levels (mol%).

4. Minor typing errors could be corrected. For example, in the abstract “SCPL” abbreviation is done two times; I also assume that some sentences are not properly written (Lines 37-40, I would suggest to remove “that”; Line 218, I wonder if I would not start with “from” rather than with “For”, …).

Response 12: We thank Reviewer 2 for his/her careful reading. The sentences have been revised. Please see the new version of manuscript.

Round 2

Reviewer 2 Report

The authors have addressed the concerns raised in round 1 of review satisfactorily. The revised manuscript can be accepted for publication in the Plants.